# Experimental Research on a Capsule Robot with Spring-Connected Legs

**DOI:** 10.3390/mi13122042

**Published:** 2022-11-22

**Authors:** Yesheng Xin, Zhen-Jun Sun, Wenjin Gu, Lei Yu

**Affiliations:** Department of Mechanical Engineering, Fujian Agriculture & Forestry University, 63 Xiyuangong Rd., Fuzhou 350002, China

**Keywords:** capsule robot, biopsy, spring, magnetic force measurement, magnetic torque

## Abstract

Based on a previous study of a novel capsule robot (CR) with spring-connected legs that could collect intestinal juice for biopsy, in this research, an experiment system is designed, and two experiments are carried out. One of the experiments measures the torque and cutting force of this CR, and the other experiment tests and evaluates the biopsy function of this CR. In the measuring experiment, we analyze how the magnetic torque exerted on this CR changes. In the experiment with a biopsy, we decompose the biopsy actions and select the most effective biopsy action. The result of the experiments shows that this CR can collect and store biopsy samples ideally, and the most effective biopsy action is the rotation with legs extended.

## 1. Introduction

Bleeding, ulcers, and tumors are common diseases in the gastrointestinal (GI) tract. However, if diagnosis and treatment are not timely, these diseases may deteriorate into serious diseases, such as cancer. Endoscopies are widely used for diagnosing GI diseases, but they are complicated to operate and require anesthesia that poses a certain risk to the cardiopulmonary system [1,2].

In 2000, the Given Imaging Company from Israel invented a wireless capsule endoscope (WCE) named “M2A”, which has been widely used in the clinical diagnosis of GI diseases since 2001. “M2A” is a capsule with a camera that moves passively via GI peristalsis, realizing the visualization of the environment in the GI tract, which is helpful for the diagnosis of GI diseases [3].

With the increasing clinical applications of capsule robots (CRs), more functions of CR are required [4,5,6,7], with the most in-demand function being the biopsy function [8]. Based on the visual information provided by the CR, the biopsy function can help doctors make a more effective and definite diagnosis of diseases [9]. In the most recent decade, many institutions have researched the biopsy function of CRs. There are mainly two areas of focus in the research of the biopsy function, one is the sampling device, and the other one is the trigger device of the biopsy.

For the sampling device, biopsy needles, biopsy razors, and biopsy forceps are widely used. Biopsy needles can take samples from deep tissue. However, the sample obtained is small because of the limited capacity of the biopsy needle. The needle may hurt or penetrate the wall of the GI tract if the needle is too long [10,11]. For example, M.C. Hoang et al. proposed a CR with a biopsy needle that could generate a 0.2 N puncture force and a 0.34 N cutting force, obtaining a biopsy sample with a size of 1 × 1.5 × 1.3 mm, but the biopsy could only be implemented once due to the limited space of biopsy needle [10]. Son et al. proposed a soft CR with a fine needle and carried out an ex vivo experiment, obtaining a biopsy sample of submucosal tumors that were located between the second and third layers of the stomach wall [11]. Biopsy razors can obtain samples from a wider surface of the GI tract than a biopsy needle or forceps, but they generally can only take the superficial tissue [12,13,14,15,16,17]. For example, Kong et al. proposed a CR with a rotational tissue-cutting razor and successfully performed a biopsy [12]. Le et al. designed a CR with a razor that successfully obtained a biopsy sample with a volume of 4.5 mm^3^ [16]. Compared to a biopsy needle, the size of samples obtained using the biopsy razor was larger but shallow. Biopsy forceps can obtain a larger sample than a biopsy needle and can obtain a deeper sample than a biopsy razor, but the biopsy forceps mainly obtain a sample by nipping and dragging, which may hurt the inner wall of the GI tract [18,19,20,21]. For example, Le et al. designed a CR with forceps, and the experiments showed that it produced a cutting force of 1.32 N (when the current of the coil was 11 A) and successfully obtained a biopsy sample with the size of 2.5 × 2 × 1 mm [21]. This force is a little dangerous to the intestine.

For the trigger device of the biopsy, magnetic control, motors, and heat-sensitive parts are mainly used. Control with the magnetic force means that a biopsy device can work without consuming energy from batteries inside the CR and has high controllability, but this requires an extra magnetic field [10,15,16,18,21,22,23]. For example, Hoang et al. controlled a CR to finish a biopsy with a magnetic field, and the puncture force and cutting force could change with the magnetic field [10]. Using the motors as a trigger device for a biopsy allows the biopsy device to start quickly and be controlled simply and reliably. However, the motor usually consumes energy from batteries inside the CR and is long [11,19]. For example, Chen et al. designed a CR with biopsy forceps driven by a motor that could generate a cutting force range from 10 to 28 N. This CR was equipped with a wireless power supply system so that there was no problem with energy, however, the direction of the biopsy was limited by the structure of this CR [19]. The heat-sensitive parts mainly include a shape memory alloy (SMA) [24] and paraffin wax, it has a variable shape so that the inner space of CR could be better arranged, however, its disadvantages are obvious, it needs energy from inner batteries to be actuated, the GI tract may be burned if the temperature was too high, and the biopsy can be performed only once due to the characteristics of heat-sensitive parts [12,13,14,17,20,25]. For example, Park et al. designed a forceps biopsy device triggered by an SMA spring, it needed 0.24 J of energy to perform a biopsy, however, it could only perform the biopsy once because of the characteristic of the SMA [20]. Control with the magnetic force means that a biopsy device can work without consuming energy from batteries inside the CR and has high controllability, but this requires an extra magnetic field [10,15,16,18,21,22,23,26,27,28,29]. According to the different sources of the magnetic field, magnetic control mainly includes two types, one is that the magnetic field is generated by permanent magnets, and the other one is that the magnetic field is generated by electromagnetic coils [30]. For example, Hoang et al. controlled a CR to finish a biopsy with a magnetic field, and the puncture force and cutting force could change with the magnetic field [10]. Kim et al. presented a method to control the magnetic force with multiple electromagnetic coils that can enhance the magnetic force and minimize the input current level to coils [28]. Guo et al. proposed a wireless modular capsule robotic system that they could use the magnetic field to move and rendezvous and separate the modules [29]. Nowadays, magnetic soft materials have made significant progress in the field of magnetic control, and they will be widely used in CR in the future [31]. Yang et al. propose an agglutinate magnetic spray that can cover inanimate objects with a film so that the objects can be magnetized and controlled by the magnetic field. Furthermore, this film can be reprogramed and disintegrated on demand, and this spray can be used to control medical robots [32].

## 2. Working Principle of CR

Intestinal diseases, such as proctitis, can cause intestinal mucosa edema, hypertrophy, becoming yellow and white, covered by mucus, and being eroded. These pathological changes can change the composition of intestinal juice, or the eroded tissues can spread into intestinal juice. Therefore, it is helpful for diagnosing intestinal diseases to collect inflammatory intestinal juice and analyze its changes or the tissues inside it. However, due to the existence of transverse and ring muscles of the large intestine, the intestine will be contracted to its center, which will cause the possible lesions to be hidden in the folds of the contracted intestine. Based on the above two reasons, this CR is designed to enlarge the intestine and collect the intestinal juice for biopsy.

### 2.1. Structure of CR

Based on previous studies of the researchers discussed above, our team proposed a novel capsule robot (CR) with spring-connected legs that can collect intestinal juice for biopsy [33].

Figure 1 shows the component parts of the CR. The CR is 16 mm in diameter and 40 mm in length. After the legs are fully expanded, the maximum distance of the tips of two opposite stretching-out legs is approximately 35 mm. The tips of every two adjacent legs are connected by a biopsy spring, and the legs are supported by three brace bars that are articulated at the nut slider. The micro-DC motor can drive the lead screw to rotate, moving the nut slider back or forth, which expands or folds the legs and the springs. The radially magnetized magnet fixed inside the CR (internal magnet) can be driven by a radially magnetized magnet outside the CR (external magnet), rotating the whole CR.

### 2.2. Principle of Expanding Intestine and Biopsy

The process of the biopsy is shown in Figure 2. After moving into the intestine (Figure 2a), the micro-DC motor can be turned on to expand the legs and springs. Then, the intestine is enlarged to expose the possible lesions hidden in the intestinal folds (Figure 2b), and the operator needs to move the external magnet to rotate the CR to collect intestinal juice (Figure 2c). Finally, the micro-DC motor is rotated in reverse to contract the legs and springs. Then the biopsy samples are collected and stored inside the springs (Figure 2d).

## 3. Measuring Experiment

### 3.1. Magnetic Torque and Measuring System

The CR is driven by a magnetic torque, which can be calculated as [34]:(1)$$T=\mu_{0}V_{IM}MHsin\varphi \;$$

In this equation, *T* is the magnetic torque of the CR that is exerted by the external magnet, *μ*_0_ is the permeability in vacuum, *V_IM_* is the volume of the internal magnet, *M* is the magnetization intensity of the permanent magnet, *H* is the magnetic field strength of the external magnet, and *φ* is the angle between $\vec{M}$ and $\vec{H}$.

According to Equation (1), the values of *μ_0_*, *V_IM_*, and *M* are constant, so the magnetic torque mainly changes with *φ* and *H*. It is difficult to control *φ* as a variable in the experiment, so we chose the angle of two magnetic poles (represented by “*θ*”) as one variable. *H* is changed with the relative position of the two magnets, and we chose the axial distance between two magnets (represented by “*D*”) as the other variable in the experiment. Because the absolute value of $\vec{H}$ is symmetrically distributed about the magnetic pole and its maximum value is on the 1/2 thickness plane of the magnet, in the experiment, it is necessary to make the 1/2 thickness plane of the two magnets coplanar.

Based on Equation (1), the measuring system should obtain the torques at different axial distances between two magnets (“*D*”) and different angles of two magnetic poles (“*θ*”). As shown in Figure 3, the system mainly includes a mechanical arm, an external magnet, an internal magnet, a measuring device, and a data acquisition device. The mechanical arm is used to move and rotate the external magnet, which is controlled by Robot Operating System (ROS) software. The core mechanism of the measuring device is a gear and rack pair, which can convert the torque into the tangential force (*F_T_*) of the gear. The data acquisition device is mainly composed of an NI industrial computer (NI PXIe-1084) and data acquisition card, Labview2019 software, and force sensor (FUTEK LSB205), which is used to measure “*F_T_*” from the gear rack.

As shown in Figure 3, the external magnet is mounted on the output shaft of the end motor of the mechanical arm with interference fit. The mechanical arm has six degrees of freedom, five of them come from the motors in the joints of the mechanical arm, and the sixth one comes from the end motor, which can adjust the position of the external magnet and control the external magnet to rotate around its own axis. Furthermore, the end motor has a photoelectric encoder so that it can control the rotation angle of the external magnet. The CR is mounted on one end of the shaft with interference fit, the gear is installed on the other end of the shaft and engaged with a gear rack, and at the right end of the gear rack is a shaft connected to a force sensor. Two shafts are connected to the left and right sides of the force sensor. The right shaft is a stepped shaft and is matched to a rolling bearing, and the left shaft is matched to a sliding bearing, which means the right shaft cannot move axially and the left shaft can only move axially for a short distance, which means that the pressure of the gear rack can be transmitted to the force sensor. For the initial status of this system, the polarities of the two magnets should be aligned, at the same time, the gear rack should be made in touch with the left shaft of the force sensor lightly so that there is no force between them. In the initial status, the indication of the sensor should be around 0 N. If we rotate the external magnet to make the CR rotate, the gear rack will press the left shaft of the force sensor, and the magnetic torque “*T*” will be converted into the tangential force “*F_T_*”. The force sensor can convert the pressure force “*F_T_*” of the gear rack into an electrical signal, which can be collected by the NI industrial computer.

As shown in Figure 4, the magnetic torque “*T*” is converted into the tangential force “*F_T_*” of the gear through the gear and rack mechanism. Therefore, the relationship between the magnetic torque *T* and the tangential force *F_T_* is:(2)$$T=F_{T}r\;$$

In this equation, *r* is the reference circle of the gear, and *r* = 7.5 mm.

**Figure 4 micromachines-13-02042-f004:**
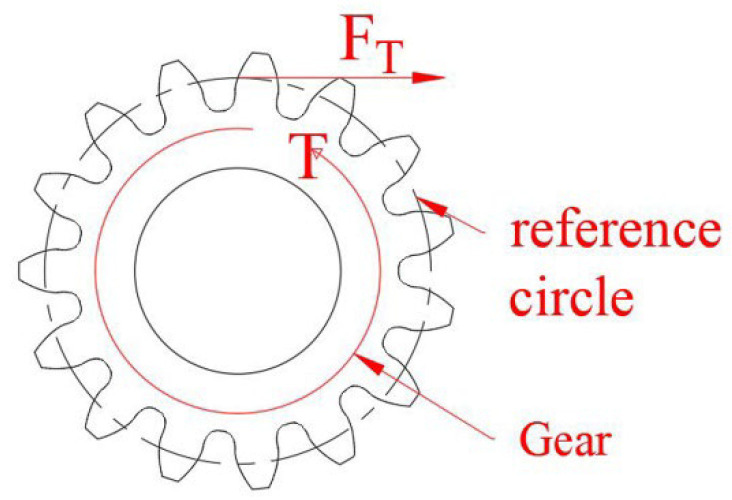
Relationship between *F_T_* and *T*.

After measuring the tangential force *F_T_*, the magnetic torque *T* can be calculated with Equation (2). Then the cutting force can be calculated as:(3)$$F_{C}=T/R\;$$

In this equation, *F_C_* is the cutting force of CR, *R* is the radius of CR and *R* = 8 mm.

### 3.2. Measuring Experiment and Result

The variable “*D*” ranges from 70 to 85 mm. When “*D*” is greater than 85 mm, the magnetic field generated by the external magnet is too weak to drive the internal magnet. Additionally, we chose 70 mm as the minimum “*D*” because there is a certain distance between human intestines and skin.

Before the experiment, the system is initialized, and the external magnet is moved by the mechanical arm to place the axis of the external magnet 85 mm above the axis of the internal magnet (at this time, *D* = 85 mm), and align the polarities of the two magnets. Then the force sensor is adjusted close to zero (the force is displayed in LABVIEW) and made to touch the end of the gear rack, with the aim of avoiding the assembly error.

The external magnet is controlled to rotate 10° every time until 180° so that the *θ* changes from 0° to 180°. Record the force (*F_T_*) from LABVIEW. Then control the mechanical arm to move down 5 mm every time until 70 mm and repeat the above steps. Figure 5 shows *F_T_* at different *D* and *θ*.

The torques (*T*) can be calculated with Equation (2), and the cutting force *F_C_* can be calculated according to Equation (3). Table 1 shows the maximums of the magnetic torque *T* and the cutting force *F_C_* at different D.

According to Equation (1), the magnetic torque between the two magnets reaches the maximum when the angle *θ* between them is 90°. In Figure 4, the maximum magnetic force is obtained when the angle *θ* is between 90° and 110°. The error between the measurement and theory may be caused by the error in the alignment of the poles of two magnets in the initial state.

## 4. Biopsy Experiment

### 4.1. Principle and Experiment System of Biopsy

Figure 6 shows the system of the biopsy experiment, which is simplified based on the measuring system. The data acquisition device and the gear and rack mechanism are removed. The fixture of the intestine is equipped, which is used to fix the intestine for biopsy.

As shown in Figure 7, in the experiment, we use a freshly isolated pig intestine to imitate a human intestine. The intestine is mounted on the fixture of the intestine, the CR (with contracted legs) is placed into the intestine through one side of the fixture, and then the micro camera is put into the intestine through the other side of the fixture to observe the CR in the intestine.

There are two purposes of the biopsy experiments:To test the effect of springs on the expansion of the folds of the intestine. In the meantime, to test whether the CR will clamp the intestine. For this purpose, we carry out the experiment of expanding the intestine.To test the biopsy effect of each action of the CR during a biopsy and the necessity of rotating the CR. For this purpose, we carry out the experiment of decomposed actions.

### 4.2. Expanding Intestine Experiment

The experiment system is assembled as shown in Figure 6. The legs of the CR are contracted, and the pig intestine is fixed in place, as shown in Figure 7. The legs of the CR are expanded, and the situations inside the intestine are observed by the micro camera that is placed on the other side of the intestine. Then the springs are removed from the CR, and the above steps are repeated. The situations inside the intestine are shown in Figure 8.

As shown in Figure 8, the CR with springs has a better effect on expanding intestinal folds, which means that the CR with springs can better collect the pathology samples that are hidden in intestinal folds. Concurrently, no intestine is clamped by the CR.

### 4.3. Experiment of Decomposed Actions

In the biopsy experiment, there are four sampling situations of the biopsy sample inside a spring. As shown in Figure 9, Situation 1 is that there is no sample in the spring, Situation 2 is that there is a small sample in the spring (less than 1/2 the volume of a spring), Situation 3 is that there are many samples in the spring (more than 1/2 the volume of a spring), and Situation 4 is that the springs are full of the samples.

We decompose the process of the biopsy and test the biopsy effect of each decomposed action. The process of biopsy can be decomposed into four actions:A.The initial status is that the legs and springs are contracted.B.The process of expanding the legs and springs.C.The process of rotating the CR while the legs are expanded.D.The process of contracting the legs and springs.

We carry out four groups of experiments, as described in this section. The steps of the experiments are shown in Table 2. The biopsy effect of each action is evaluated by the proportions of the above four situations (as shown in Figure 9) among six springs.

Group I of the experiments is intended to observe the biopsy effect of action A (initial status) and whether the CR can collect the intestinal juice while it is moving in the intestine without expanding the springs. The result of Experiment I is shown in Figure 10.

As shown in Figure 10, in Experiment 1 of this group, four springs are in Situation 1 and two springs are in Situation 2. Among the 10 experiments of Group I, the total number of springs that are in situation 1 is 39, and the total number of springs that are in situation 2 is 21. Therefore, in the first group of experiments, the proportion of springs that are in Situation 1 is 65%, and the proportion of springs that are in Situation 2 is 35%, which means that a small amount of intestinal juice can be collected while the CR is moving with springs contracted, but these biopsy samples are mainly accidentally collected at the ends of the springs. Therefore, when the biopsy springs are contracted, the CR will neither perform an effective biopsy nor accidentally collect large quantities of unwanted samples.

Group II of the experiments is intended to observe the biopsy effect of action B. The result is shown in Figure 11.

As Figure 11 shows, the sampling situations of the springs are mainly in Situation 2 (96.67%), which means that small samples can be collected by increasing the standing time, the reason for which is that the intestinal juice is supported by surface tension. However, if there is relative movement between the spring and the intestine, more intestinal juice will be collected, and Experiment III and IV proved it.

Group III of the experiments is to observe the biopsy effect of action D that the CR does not rotate. The result is shown in Figure 12.

As Figure 12 shows, the sampling situation of the springs is mainly in Situation 3 (66.67%) and Situation 4 (25%), which means that the CR can collect a certain amount of biopsy sample through the action of expanding and contracting the legs, but the springs are not fully filled.

Group IV of the experiments is intended to observe the biopsy effect of action C (rotating) by forming a controlled experiment with Experiment III. The result is shown in Figure 13.

As Figure 13 shows, the springs are mainly in Situation 4 (98.33%), which means that the springs are fully filled with intestine juice. In contrast to Experiment III, the rotation of the CR while the legs are fully expanded is necessary and has a significantly better biopsy effect.

In summary, of the four sets of experiments, the most effective biopsy process is to expand the legs and springs of CR from the initial status, rotate the CR for a round by rotating the external magnet, and then contract the legs and springs. Nearly all of the six springs can be fully filled with biopsy samples through this biopsy process.

## 5. Conclusions

In this research, we designed an experiment system for a CR with spring-connected legs. This experiment system was used for measuring the torque and cutting force and testing the biopsy effect of the CR.We measured the torques and cutting force during a biopsy of this CR. When the axial distance between the external magnet and the internal magnet (*D*) was 70 mm, the torque exerted on the CR reached 1.620 N·mm at most, at this time, the cutting force of the CR was 0.203 N.We carried out several experiments to research the biopsy function of this CR. The biopsy effect of each decomposed action of the biopsy was tested. The most effective action was the rotation of the CR while the legs were fully expanded. Based on this experiment, we designed the most effective process for a biopsy.

## 6. Discussions of Future Work

In the future, the work of this CR will mainly focus on the following aspects:To reduce the pollution of the collected biopsy sample while the CR is moving inside the intestine, the capsule shell will be optimized so that the springs can be embedded in the shell.To save the internal space of the CR, the interior of the CR will be optimized. The micro-motor will be replaced by an internal magnet, which will cooperate with external magnets to expand the legs.In addition to the biopsy function, the drug-delivery function will be researched, and we will store medicine inside the springs and test whether the medicine can be released to a lesion.

## Figures and Tables

**Figure 1 micromachines-13-02042-f001:**
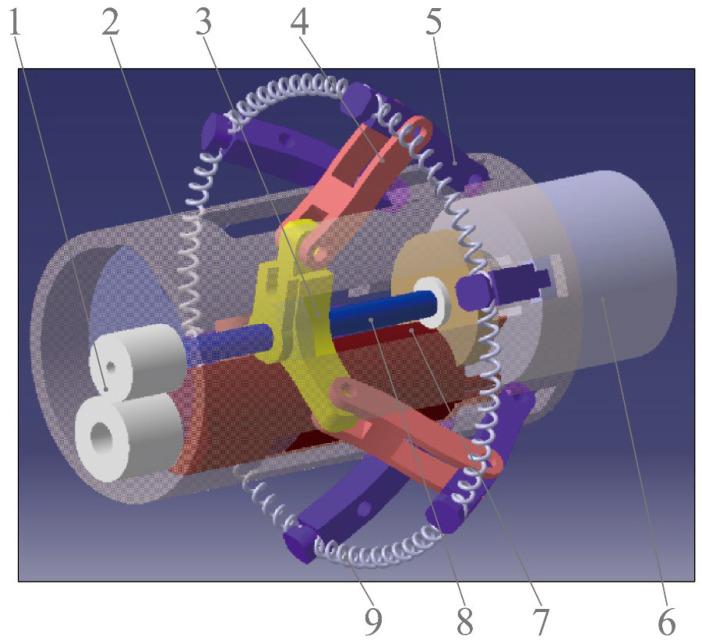
Structure of capsule robot. 1. Gear pair, 2. Shell, 3. Nut slider, 4. Brace bar, 5. Leg, 6. Radially magnetized magnet, 7. Micro-DC, motor 8. Lead screw, 9. Spring.

**Figure 2 micromachines-13-02042-f002:**
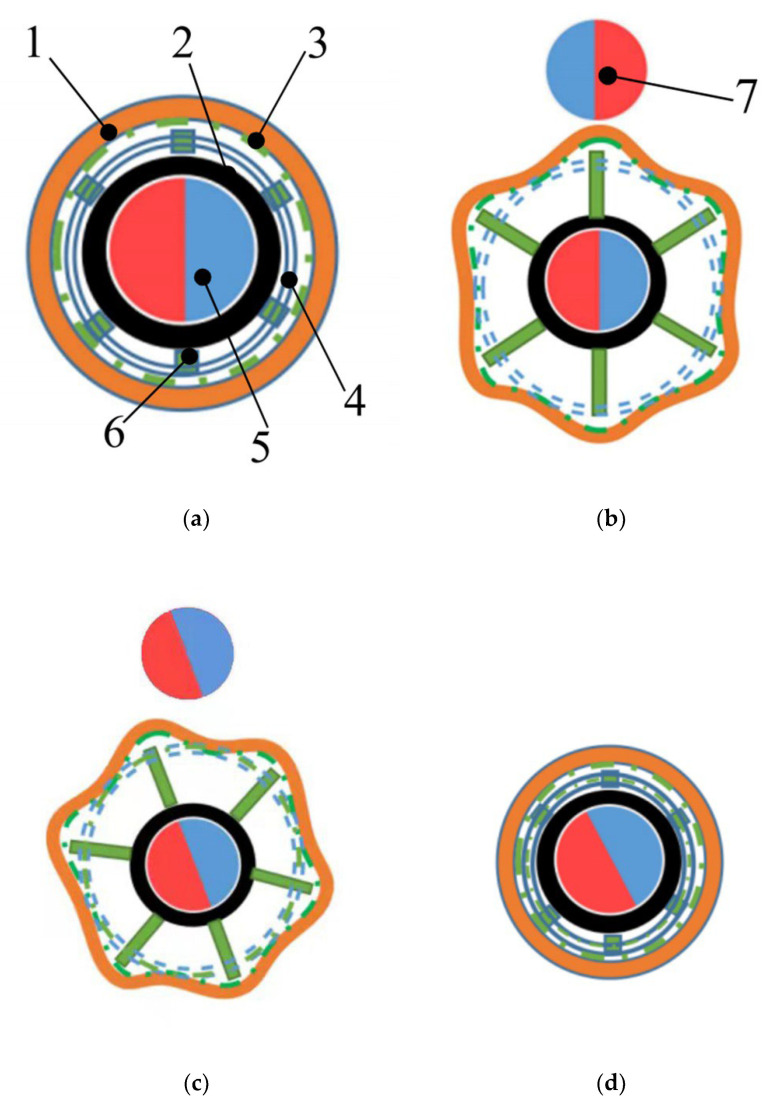
Biopsy process. (**a**) Initial status, (**b**) expanding legs, (**c**) rotating CR, (**d**) contracting legs. 1. Intestine, 2. Capsule shell, 3. Intestinal fluid, 4. Spring, 5. Internal radially magnetized magnet, 6. Leg, 7. External radially magnetized magnet.

**Figure 3 micromachines-13-02042-f003:**
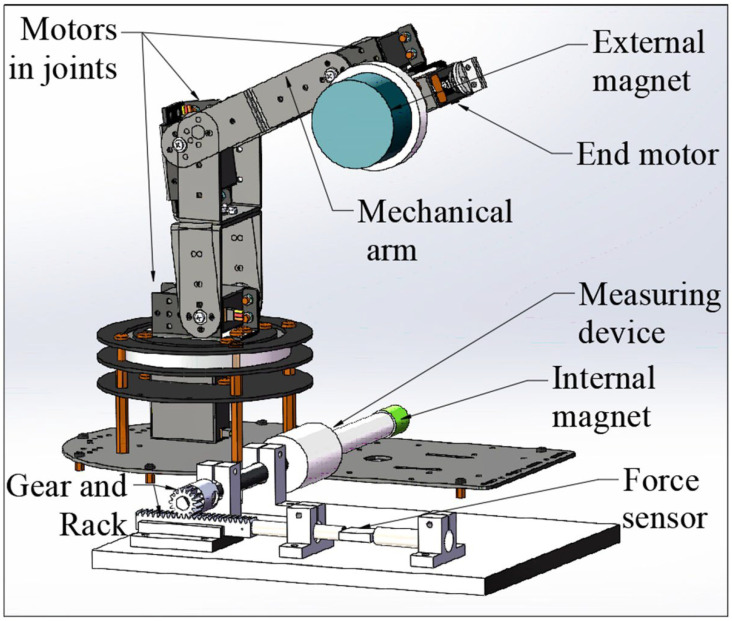
Magnetic torque measuring system.

**Figure 5 micromachines-13-02042-f005:**
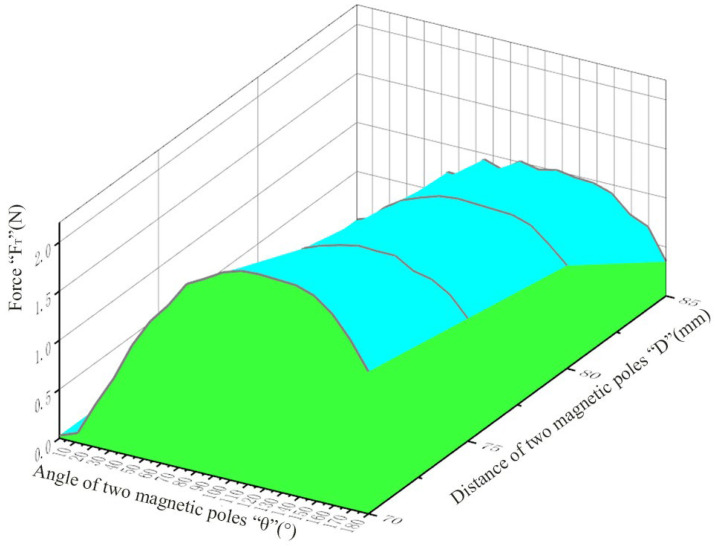
Forces at different *D* and *θ*.

**Figure 6 micromachines-13-02042-f006:**
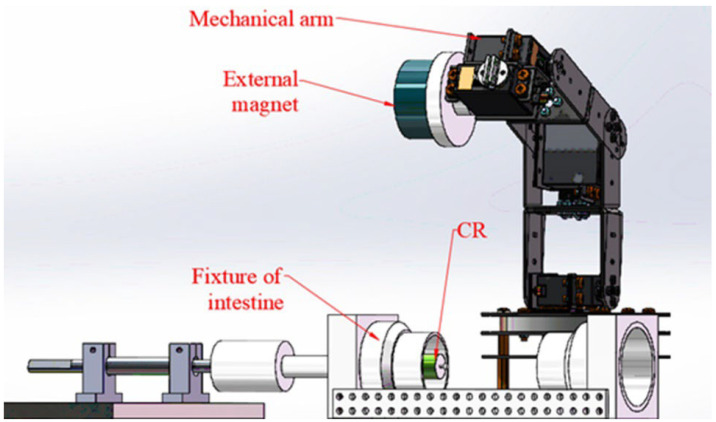
Biopsy experiment system.

**Figure 7 micromachines-13-02042-f007:**
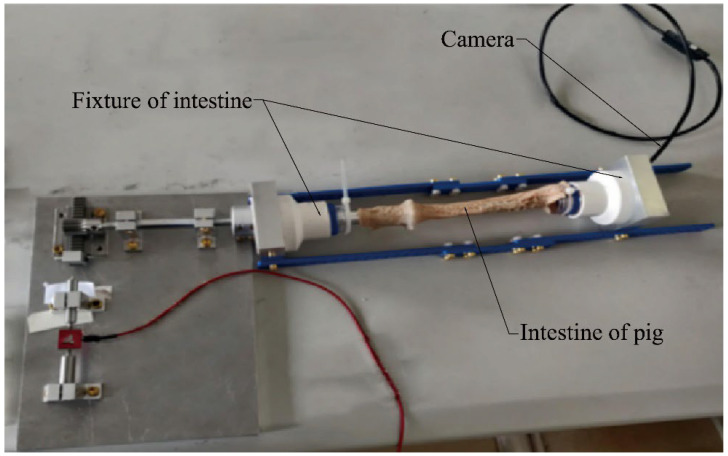
Use of the fixture of intestine.

**Figure 8 micromachines-13-02042-f008:**
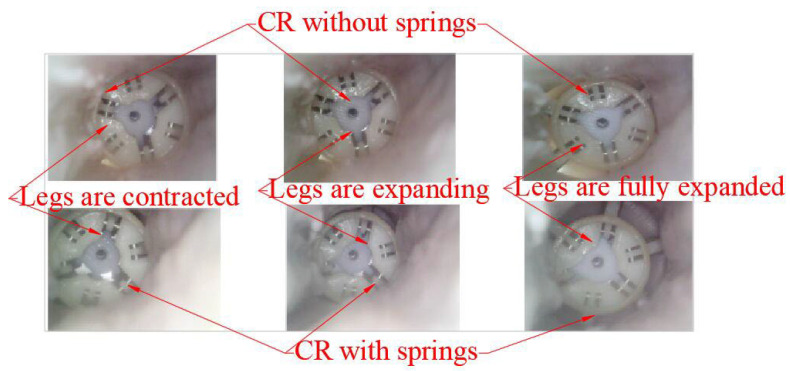
Comparison of the effects of the CR on the expansion of intestinal folds (with or without springs).

**Figure 9 micromachines-13-02042-f009:**
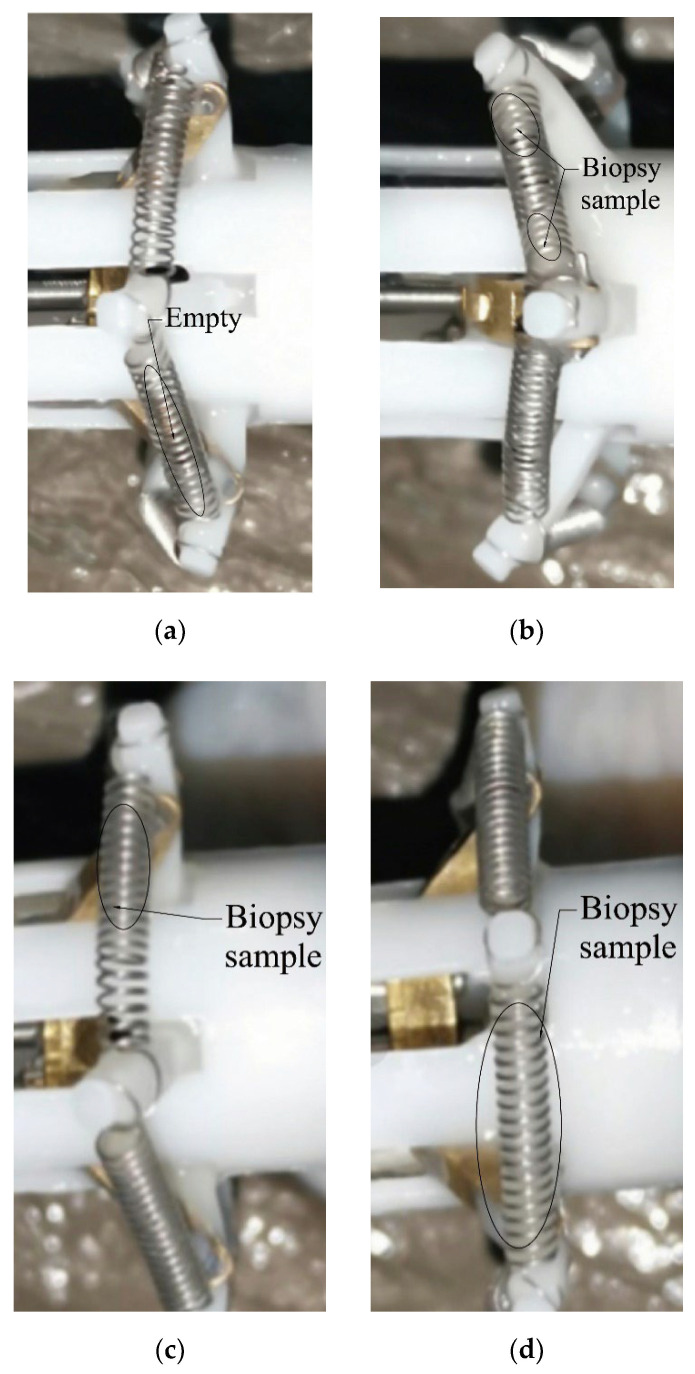
Situations of the biopsy sample inside a spring. (**a**) Situation 1, (**b**) Situation 2, (**c**) Situation 3, (**d**) Situation 4.

**Figure 10 micromachines-13-02042-f010:**
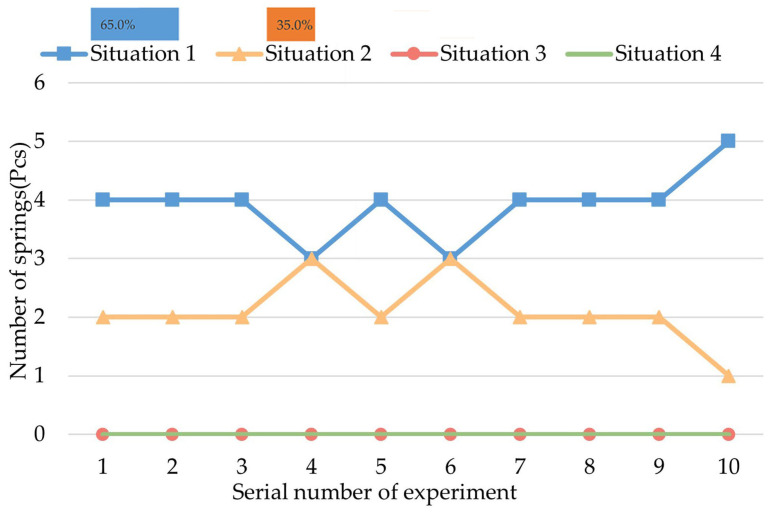
Result of group I of experiments.

**Figure 11 micromachines-13-02042-f011:**
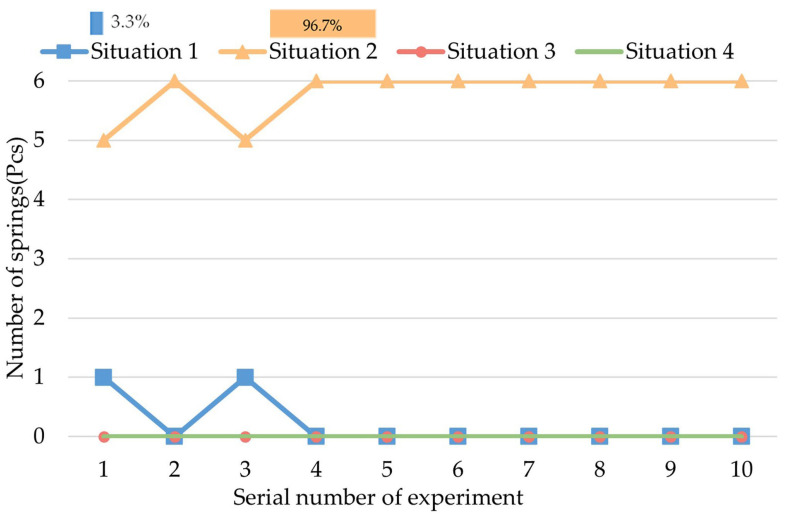
Result of Group II of experiments.

**Figure 12 micromachines-13-02042-f012:**
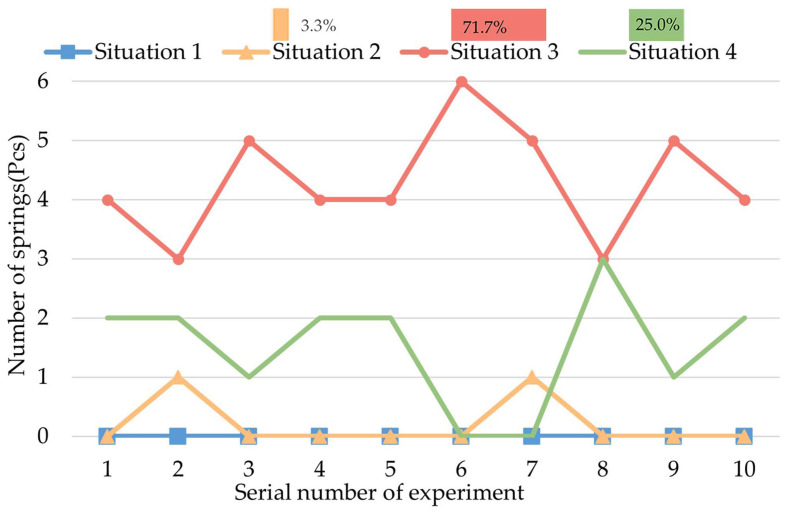
Result of Group III of the experiments.

**Figure 13 micromachines-13-02042-f013:**
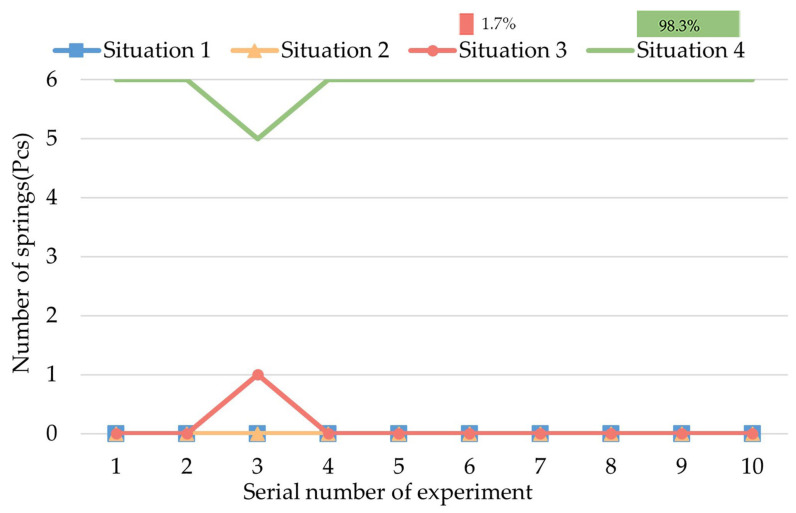
Result of Group IV of experiments.

**Table 1 micromachines-13-02042-t001:** T and *F_C_* at different *D* from measurement.

*D*/mm	*T*/N·mm	*F_C_*/N
70	1.62	2.16
75	1.29	1.72
80	1.095	1.46
85	0.758	1.01

**Table 2 micromachines-13-02042-t002:** Steps of the experiments.

Group Number *	Preparation Step	Intermediate Step	Last Step
I	Set CR in the initial status (legs are contracted), cover the CR with the intestine	Move the intestine back and forth to simulate the movement of the CR inside the intestine.	Remove the intestine carefully and expand the legs to observe and record the sampling situation of the springs.
II		Fully expand the legs and stand for 30 s (increase the standing time by 30 s each time).	
III		Expand and then contract the legs.	
IV		Expand the legs, rotate CR for a round, and contract the legs.	

* Each group of experiments includes 10 experiments.

## Data Availability

Not applicable.

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
