# Peer review of "Experimental Research on a Capsule Robot with Spring-Connected Legs"

_micromachines, 2022, doi:10.3390/mi13122042_

Round 1
Reviewer 1 Report
The paper designed designed an experiment system and carried out two experiments: one is to measure the torque and cutting force of the capsule robot proposed previously, the other is to test and evaluate the biopsy function of the robot. The presented work is more like an experimental report. Its innovation is not enough to be accepted by the journal, but to resubmit some international conferences, such as ICRA, IROS.
There are some suggestions as followings.
(1) The paper aims to design a device for measuring the characteristic of CR. I think the authors should describe more about the dexterity of mechanism design or sensors that just solves the magnetic measure problem, than the detailed operation process.
(2) Mentioning the detailed experimental process, too much description contents are repeated. For example, “The steps of experiment are: Set the CR in initial status, cover the fresh isolated pig’s intestine o….” in Experiment I, p9, also in Exp.II, p10,…Exp. IV, p11.
The repeat taking up too much lines is not necessary for the paper’s key ideas.
(3) The device in Fig.7 should be a prototype of 3D picture in Fig.6.
(4) The measurement of magnetic torque and cutting force may be really important for the designed CR. Authors should mark the force sensor in the Fig.7. Another, authors sample a tangential force when the magnet rotating 10 degree. Here, the paper should discuss the influence of the velocity curve or acceleration of the external magnet on the tangential force. It is because the permanent magnet is a gradient magnetic field. Even if the thet=0, only if the external magnet rotates, the internal magnet will rotate. The thet in Eq.(1) maybe not the sample interval thet=10.
(5) In table 1, why did authors choose the distance of two magnetic poles from 90 to 70mm? It is because the magnetic drive force is too little to drive the CR when the distance >90mm, is it? Please discuss it.
(6) I wonder if Situation I to 4 are 4 actions in Fig.10-Fig.13. If it is yes, authors would better use the same word.
(7) For illustrating Fig.10-Fig.13, authors note that” the springs are mainly in situation 1(65%) and situation (35%)”, just like these proportion values. I want to know how to calculate these values using which equations. For example, 65% in Fig.10, 96,67% in Fgi.11, etc.
Reviewer 2 Report
See attached.

Round 2
Reviewer 1 Report
The manuscript is improved a lot in the version. For the endoscopic capsule, besides its novel design, it necessarily exploits magnetic function for dexterous actions of legs. I think other magnetic methods for small-scale robots need to be mentioned in the paper, such as different movement actuated by magnets (Science Robotics, 2020, DOI: 10.1126/scirobotics.abc8191; IEEE Transactions on Industrial Electronics ,2021, DOI: 10.1109/TIE.2021.3120443). Another tip is the most of the references cited in the manuscript are not new. It is suggested the 2/3 references were better published within 5 years.
Reviewer 2 Report
The number of sections should start from 1, instead of 0.
